# Barium bioaccumulation by bacterial biofilms and implications for Ba cycling and use of Ba proxies

Francisca Martinez-Ruiz [1], Fadwa Jroundi[2], Adina Paytan[3], Isabel Guerra-Tschuschke[4], María del Mar Abad[4] & María Teresa González-Muñoz[2]

Ba proxies have been broadly used to reconstruct past oceanic export production. However, the precise mechanisms underlying barite precipitation in undersaturated seawater are not known. The link between bacterial production and particulate Ba in the ocean suggests that bacteria may play a role. Here we show that under experimental conditions marine bacterial biofilms, particularly extracellular polymeric substances (EPS), are capable of bioaccumulating Ba, providing adequate conditions for barite precipitation. An amorphous P-rich phase is formed at the initial stages of Ba bioaccumulation, which evolves into barite crystals. This supports that in high productivity regions where large amounts of organic matter are subjected to bacterial degradation, the abundant EPS would serve to bind the necessary Ba and form nucleation sites leading to barite precipitation. This also provides new insights into barite precipitation and opens an exciting field to explore the role of EPS in mineral precipitation in the ocean.

[1] Instituto Andaluz de Ciencias de la Tierra (CSIC-UGR), Avda. de las Palmeras 4, 18100 Granada, Armilla, Spain. [2] Department of Microbiology, Faculty of Science, University of Granada, Campus Fuentenueva, 18002 Granada, Spain. [3] Institute of Marine Sciences, University of California Santa Cruz, Santa Cruz, CA 95064, USA. [4] Centro de Instrumentación Científica (CIC), University of Granada, Campus Fuentenueva, 18071 Granada, Spain. Correspondence and requests for materials should be addressed to F.M-R. (email: fmruiz@ugr.es)

The biological pump plays a crucial role in ocean chemistry and the global carbon cycle[1]. Understanding past variations in the efficiency of the biological pump calls for the reconstruction of ocean export productivity, which fluctuates considerably over multiple time scales[2]. As there is a strong link between particulate organic carbon (POC) and particulate Ba fluxes in the ocean[3], Ba proxies have been used to reconstruct carbon fluxes and past marine export production[4]. Nevertheless, a comprehensive understanding of nutrient cycling and export production in past oceans, including the significance of the biological pump and its efficiency, are hindered by our lack of full understanding of Ba cycling in the ocean and controls on barite ($BaSO_4$) formation in the water column. It is well known that barite precipitates in undersaturated seawater as micron-sized crystals in association with sinking biogenic debris[5], and its nutrient-like behavior and link to carbon remineralization at depth have been widely demonstrated[3,4,6]. However, the processes that lead to precipitation of barite in microenvironments within sinking particulate matter are not well constrained. It has been suggested that bacteria may play a role, since observations in the ocean water column support a close relation between bacterial activity and particulate non-lithogenic Ba in the water column[7–9]. It has also been demonstrated that under experimental conditions diverse marine bacteria from different habitats and of phylogenetically diverse species can promote barite precipitation[10,11]. Seawater Ba isotopic composition likewise points to a biological role in barite precipitation. This process preferentially incorporates the lighter isotopes of Ba with larger fractionation factors than for inorganic barite, and the residual waters remain isotopically heavier[12–14].

Although the microbial role in precipitation of sulfates has been poorly investigated in comparison to the microbial involvement in carbonates formation, microbial barite precipitation has been described in hot springs environments, where bacteria play a role either in oxidizing sulfur compounds to generate sulfate or in accumulating Ba in microbial mats[15,16]. In marine environments, bacterial involvement in barite formation has recently been reported for a cold seep where barite precipitated on filaments of sulfide-oxidizing bacteria. Laboratory experiments with isolates of these sulfide-oxidizing bacteria also show that under low sulfate conditions the sulfate generated by sulfide-oxidizing bacteria fosters rapid barite precipitation localized on the cell biomass[17]. In most previous experiments, bacteria provided the necessary sulfate for barite precipitation. However, this is not analogous to the process occurring in the open ocean, where sulfate concentrations in seawater are high; hence providing sulfate does not represent the role of bacteria in marine microenvironments. Given that ocean seawater is generally undersaturated with respect to barite due to low concentrations of Ba, the formation of barite depends upon a mechanism that serves to increase Ba concentrations at the sites of barite precipitation. Bioconcentration of Ba by phytoplankton has been reported through culture experiments and Ba isotope-spiking experiments[18]. Bearing in mind that the decay of phytoplankton is bacterially mediated, bacterial productivity would increase along with the availability of organic matter substrate and in turn with export productivity.

In this context, we explore the potential of certain marine bacteria to bioaccumulate Ba in biofilms so as to shed light on the processes allowing marine bacteria in their natural habitat to promote barite precipitation. Although past experimental work and the close link between bacterial production and particulate Ba abundance in seawater suggest a link between bacterial activity and barite formation, the precise mechanisms for precipitation and the specific role of marine bacteria in the process have not been investigated. The capability of certain marine bacteria to provide sulfate in culture experiments promoting barite precipitation had served to demonstrate the microbial role in barite precipitation in low sulfate settings. However, this role is underplayed in the sulfate rich seawater. We show here that Ba bioaccumulation on bacterially produced biofilms is the crucial step for barite formation. We also show how barite forms under experimental conditions from an amorphous precursor that evolves to barite crystal, and speculate that a similar process occurs in the ocean.

## Results

**Bacterial cultures**. Three marine bacterial strains were selected for this study: *Idiomarina loihiensis* MAH1, *Marinobacter hydrocarbonoclasticus*, and *Planomicrobium okeanokoites* (references CECT-5996, DSM-50418, and DSM-15489, respectively). These strains were chosen because they belong to three phylogenetically different genera widely distributed in the marine environment. The first two are Gram-negative bacteria ($\gamma$−Proteobacteria) and the third is a Gram-positive bacterium (Firmicutes). Bacteria grown in a liquid medium and washed with NaCl solution were suspended in synthetic seawater without sulfate, allowing biofilm formation. Two concentrations of $BaCl_2$ in the media (2 and 20 mM) were used.

**Scanning electron microscopy analysis**. Scanning electron microscopy (SEM) observations have demonstrated that Ba accumulated in the biofilms produced by all three marine bacteria in both media (2 and 20 mM Ba) after 24 h. Figures 1 and 2 show the Ba accumulated in bacterial biofilms and the composition of the precipitates. Energy-dispersive X-ray (EDX) spectra show the elements bound to extracellular polymeric substances (EPS); elements derived from the glass coverslips used for the experiments (e.g. Si, K) are shown in a background spectrum (Supplementary Fig. 1). The composition of the precipitates reveals that a P-rich phase formed at the initial stages of Ba bioaccumulation (Fig. 1a, b) and this phase later evolve into barite (Fig. 1d, e).

**High-resolution transmission electron microscopy**. High-resolution transmission electron microscopy (HRTEM) analyses allowed the precise composition of the Ba precipitates to be determined; EDX maps and spectra show the elemental distributions of Ba, P, and S in the biofilms, and the presence of polyphosphate grains enriched in Ba in some cells (Figs. 3 and 4; Supplementary Figs. 2, 3, 4). These results further demonstrate that Ba is initially associated with P in the biofilms, and also that these initial P-rich phases are amorphous or very poorly crystallized, as evidenced by the selected area electron diffraction (SAED) images that provide evidence of amorphous or very poorly crystallized precipitates. In contrast, particles with higher S content, which developed at later stages of the experiment, are more crystalline. SAED images (Fig. 4b) obtained from S-rich precipitates show the crystalline nature of these phases. The obtained d-spaces for these crystals present a deviation from those of pure barite crystals, as the precipitates are still relatively rich in P. A wide range of compositions was observed, from initial P-rich precipitates to intermediate phases with mixed P and S and the almost complete substitution of phosphate by sulfate, with the subsequent formation of barite crystals.

**Composition of natural barite from seawater**. Barite particles from sinking particulate matter collected using a multiple-unit large-volume in situ filtration system (MULVFS) from the ocean water column (North Atlantic, Knorr cruise 98 sample WCR 82 H, September 1982) were analyzed under SEM for comparison to

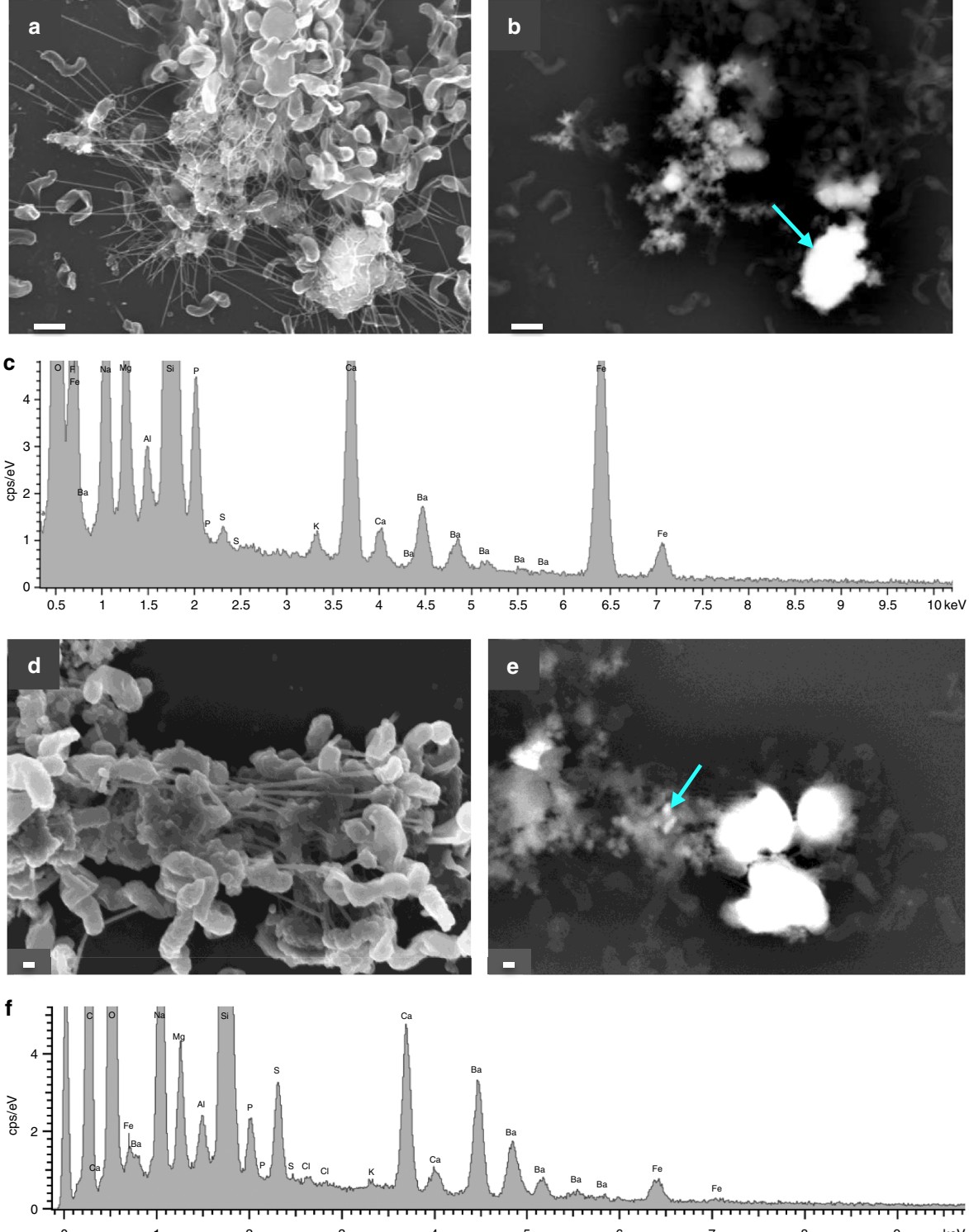

**Fig. 1** SEM images of cells and EPS from *l. loihiensis* and the bioaccumulated Ba. **a** Cells, EPS, and Ba precipitates after 48 h of incubation in SSW2 medium. The image was obtained by secondary electrons with the Inlens detector at 5 kV. **b** Same image shown in **a** in backscattered electron (BSE) mode by AsB detector at 20 kV. Scale bar in **a** and **b** represents 300 nm. **c** Representative SEM-EDX spectrum (analyzed spot marked with an arrow in BSE image), in which the vertical scale is enlarged to show the lower intensity of S peak in relation to P. Elements, other than those bound to EPS, derive from the glass coverslips used in the experiments (e.g. Si, K), see Supplementary Fig 1 for a background spectrum. **d**, **e** *l. loihiensis* MAH1 after 96 h of incubation, and the same conditions previously mentioned for photographs **a** and **b**. **f** Spectrum showing the enhanced intensity of the S peak in relation to P. Scale bar represents 200 nm. Photographs **d** and **e** and corresponding spectrum were obtained with a Zeiss SUPRA40VP and the rest with an AURIGA FIB-FESEM Carl Zeiss SMT

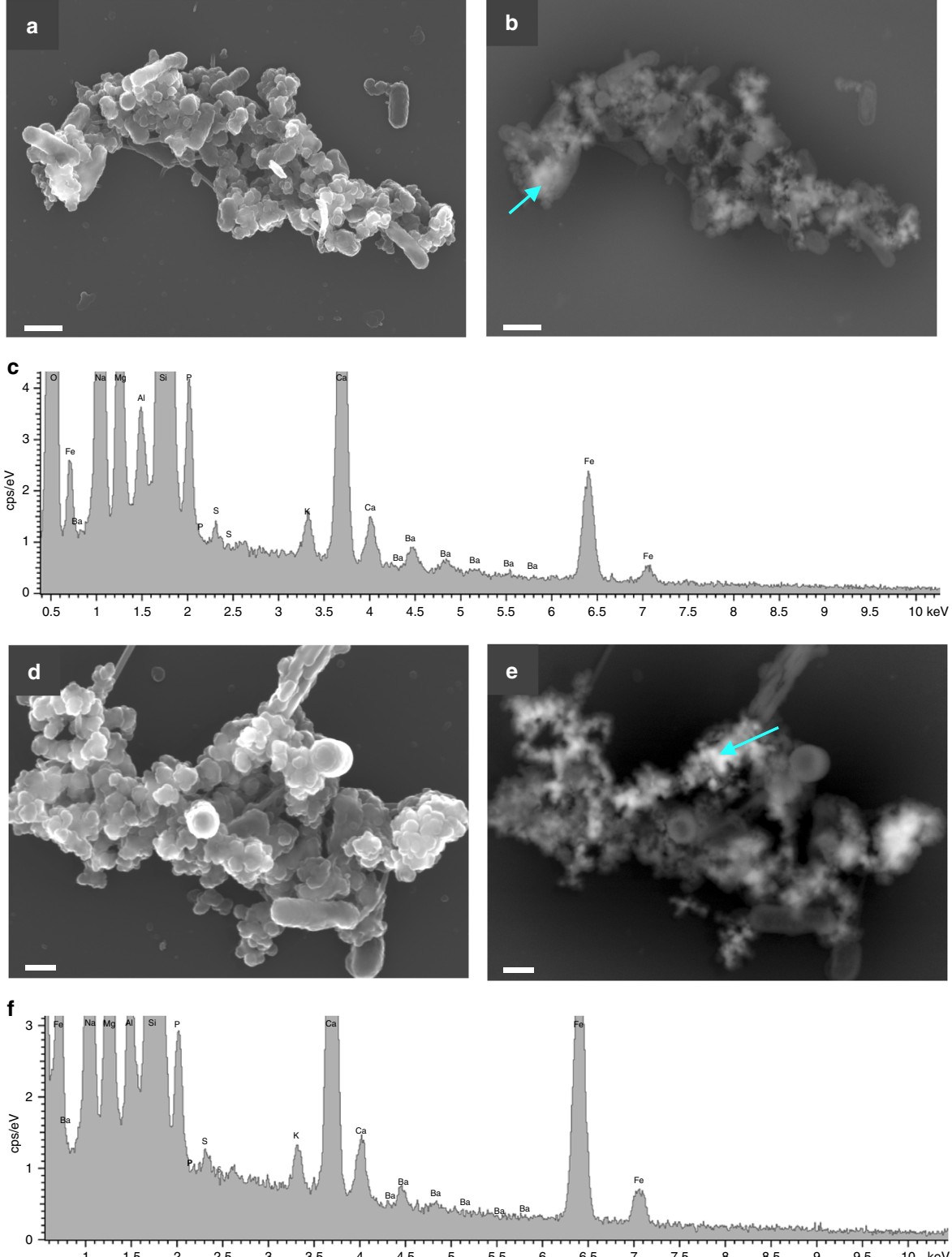

**Fig. 2** SEM images of *M. hydrocarbonoclasticus* and *P. okeanokoites* bacteria. **a**, **b** Photographs of *M. hydrocarbonoclasticus* obtained by the Inlens detector and backscattered electron mode, respectively. Scale bar represents 1 μm. **c** Representative SEM-EDX spectrum. Incubation time was 48 h and the medium SSW2. The corresponding spectrum also shows the composition of the Ba-rich aggregates. **d**, **e** Photograph of *P. okeanokoites* after 96 h of incubation in SSW2 medium obtained under the same microscopy conditions. Scale bar represents 200 nm. **f** Composition of the precipitates shown in **d** and **e** images

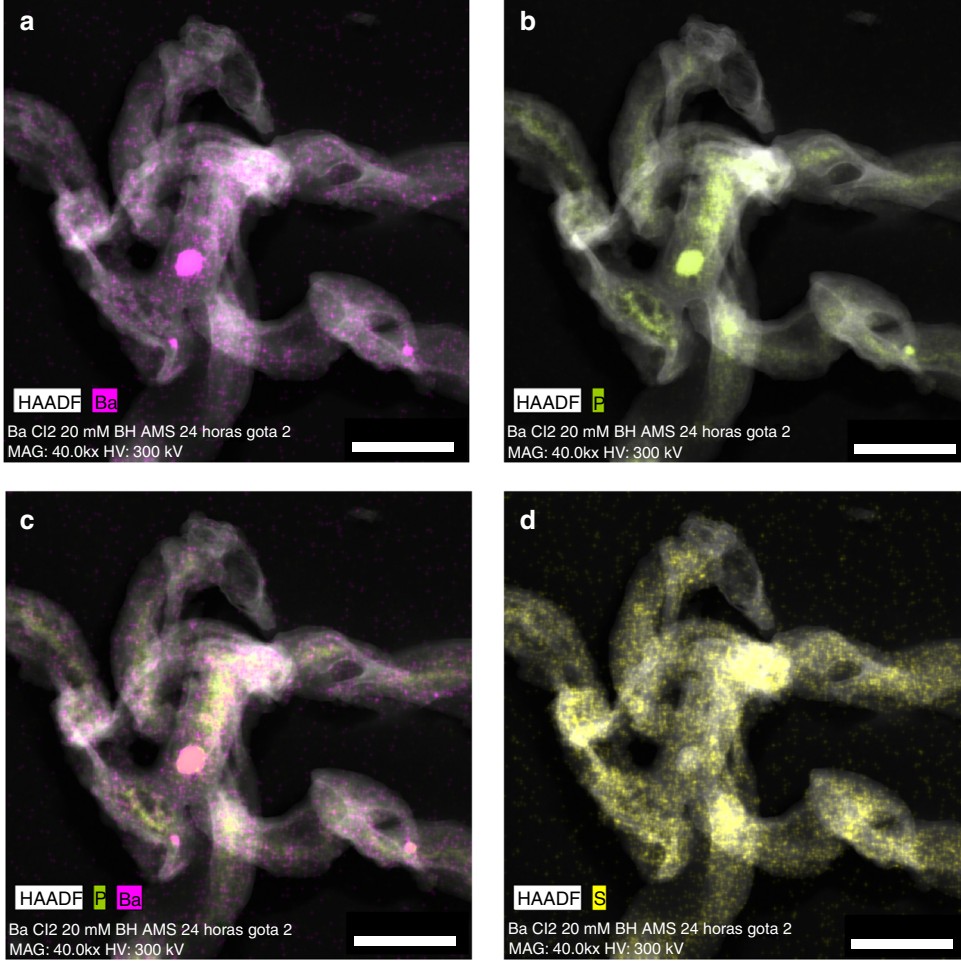

**Fig. 3** HRTEM EDX maps of Ba, P, and S in bacterial cells. **a**, **b** High-angle annular dark field (HAADF) STEM images and corresponding EDX maps showing the distribution of Ba and P respectively in bacterial biofilms of *I. loihiensis* MAH1 after 48 h of incubation in SSW2 medium. **c** Combined map of Ba and P showing the similar distribution of these elements . **d** Map of S showing the scattered distribution of this element within bacterial cells. Scale bar represents 400 nm

the experimental precipitates. Similar to our experiments, in the natural samples, P was also localized with Ba, particularly in very tiny grains (Fig. 5), while in larger grains P is not noticeable by EDX analyses.

## Discussion

The importance of marine barite and Ba proxies for reconstructing past ocean export productivity has been widely demonstrated[4]. The precipitation of barite in undersaturated seawater[19] strongly suggests that biogeochemical processes rather than purely abiotic processes must be involved in such precipitation. Although no explicit mechanism has been identified for the nucleation and growth of barite crystal in the ocean water column, diverse lines of evidence suggest that bacteria play a major role. Dehairs et al.[7] investigated particulate Ba content in the North Pacific and show that the vertical distribution of particulate non-lithogenic Ba in the water column (composed primarily from barite) closely follows trends in bacterial activity and oxygen consumption. In the Australian sector of the Southern Ocean, Jacquet et al.[8] observed that mesopelagic Ba content is correlated with bacterial activity, with an increasing abundance of particulate Ba when bacterial activity is high. Similarly, Planchon et al.[9] show that mesopelagic particulate Ba distribution reflects

bacterial degradation of organic matter, and that this is related to oxygen consumption and bacterial carbon respiration in the Atlantic sector of the Southern Ocean. Experimental work also demonstrated that phytoplankton may play a crucial role in bioconcentrating Ba[18], and since sinking organic matter serves as a substrate for heterotrophic bacteria, the link between bacterial abundance and export productivity may be a key factor for Ba bioaccumulation in microenvironments in the ocean water column. Indeed, in natural environments, Ba enrichment associated with microbial biomass has been described in diverse settings, including thermal springs[15,16,20], biofilms in the Roman Catacombs[21], bacterial EPS at fumaroles in Solfatara Crater, Italy[22], and in filaments of sulfide-oxidizing bacteria in marine cold seep[17]. Likewise, culture experiments demonstrated the capability of bacteria to promote barite precipitation under laboratory conditions[10,11,23]. In such experiments, bacterial metabolism supplied the necessary sulfate for barite formation. Our experimental work also provided new insights into precipitation mechanisms, showing that phosphate groups are the main ligand for Ba binding and that phosphate is eventually substituted by sulfate to form barite. The binding of many elements to phosphate groups is a common biomineralization process leading to microbially induced mineral precipitation. For example,

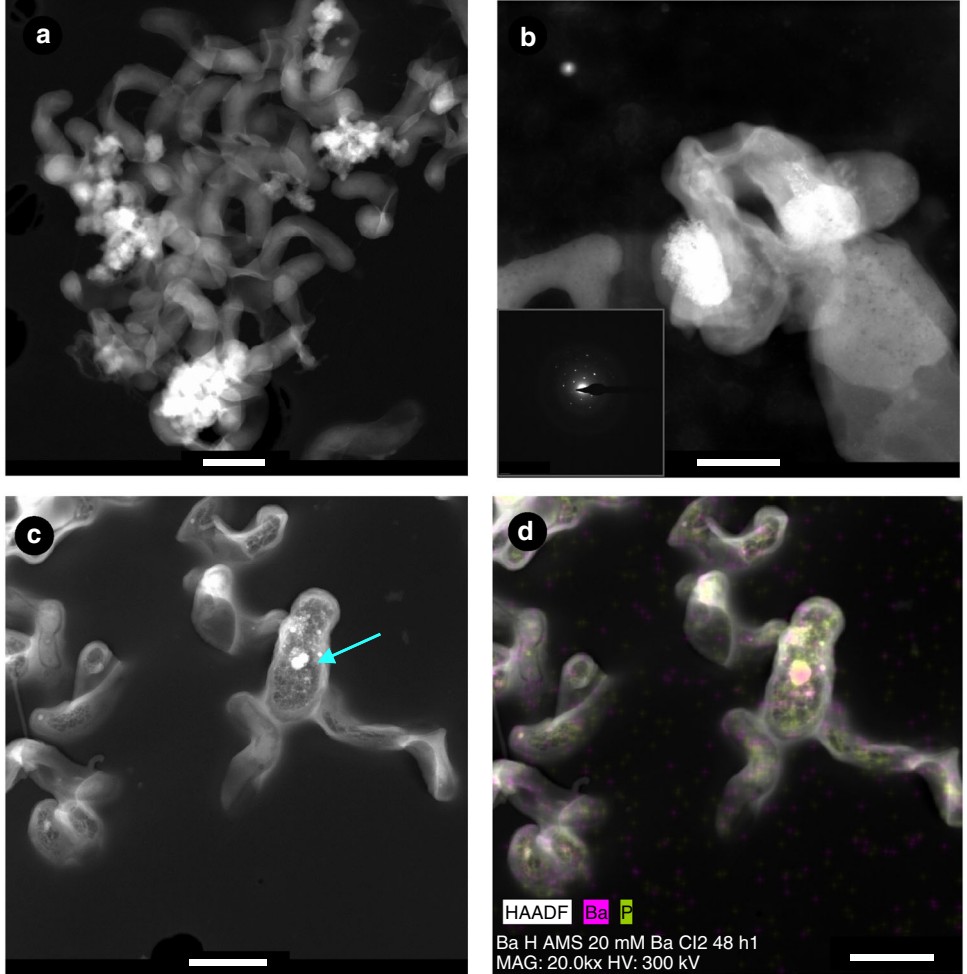

**Fig. 4** HRTEM images of bacterial biofilms. **a** HAADF images showing bacterial biofilms of *I. loihiensis* MAH1. **b** High-magnification photograph of precipitates formed after 48 h of incubation in SSW2 medium. A selected area electron diffraction image is included in photograph **b** to show the crystalline nature of the precipitates (the composition is shown in Supplementary Fig. 2). **c** Intracellular grains enriched in P (indicated with an arrow). **d** Corresponding EDX map showing the Ba and P distribution (spectrum is provided in Supplementary Fig. 3). Scale bar represents: **a** 500 nm, **b** 200 nm, **c** 500 nm, **d** 700 nm

Rivadeneyra et al.[24] studied aragonite precipitation by *Chromo-halobacter marismortui* and demonstrated that, over time, the composition of bioliths changed from amorphous phases of Ca phosphate and carbonate to precipitates made essentially of aragonite and struvite. Furthermore, the ability of polyphosphates to chelate metal ions has been widely demonstrated. In fact, the intracellular chelation of heavy metals by polyphosphate may serve to reduce heavy metal toxicity and improve cellular toler-ance to metals[25]. We surmised that, in seawater, phosphate groups on bacterial biofilms could play a major role in con-centrating Ba, and act as nucleation sites for barite formation. Although phosphate and sulfate groups have different sizes (the phosphate group being larger), the mutual substitution of sulfate and phosphate groups is a common process in nature, as are solid solutions between phosphate and sulfate phases[26]. The adsorption of divalent cations on EPS and cell walls has been widely demonstrated[15,27]. Indeed, microbial mats and biofilms are commonly enriched in calcium and magnesium as compared to surrounding waters[28]. Trace metal incorporation by bacteria has been demonstrated for other metals as well (e.g. U[29]) specifically in the case of marine bacteria used in this study such as *I. loihiensis*.

Our experimental work represents a step forward in this field and demonstrates that common marine bacteria (*I. loihiensis*, *M. hydrocarbonoclasticus*, and *P. okeanokoites*) can bioaccumu-late Ba, which is essential for increasing Ba concentration and establishing nucleation sites in microenvironments in which barite precipitates. These results lead one to further hypothesize that barite precipitation in the ocean water column is mediated by Ba enrichment, initially bound to phosphate groups in biofilms and other organic substrates, and that eventually the phosphate is substituted by sulfate, which is readily available in seawater, to form crystalline barite. While bacteria are known to play a role in providing sulfate for barite precipitation in certain natural environments such as sulfidic springs[15,17,20], in the ocean water column, where sulfate is abundant, the bacterial role would consist primarily of providing biofilms that bind Ba as the first step for barite precipitation. The bacteria would thus play an indirect role in the precipitation process, providing binding sites on EPS, cell walls, or in polyphosphate granules within the cells where Ba can bind to phosphate, followed by substitution of phosphate with sulfate to form barite. The observation of amor-phous phosphate precursor associated with Ba in natural sinking particulate matter collected using a MULVFS reinforces the

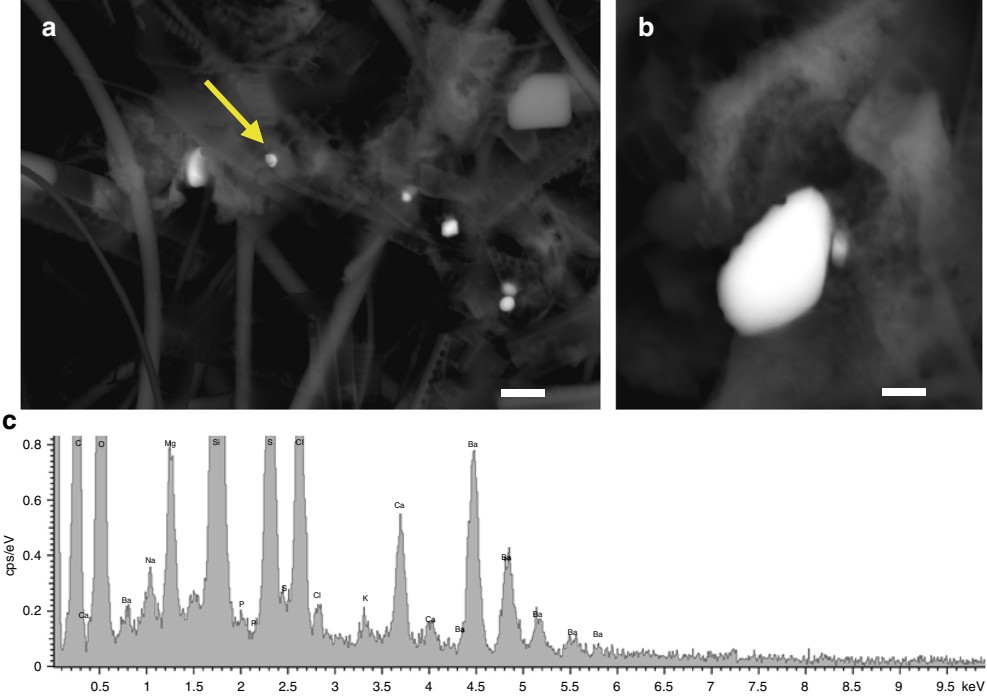

**Fig. 5** SEM images and composition of natural barite from the ocean. **a**, **b** SEM micrographs showing examples of marine barite particles from the ocean water column (North Atlantic) in sinking particulate matter collected using a multiple-unit large-volume in situ filtration system. Scale bar represents 3 µm in **a** and 1 µm in **b**. **c** Representative spectrum showing the composition of barite particles. The spectrum corresponds to the particle indicated with the arrow in photograph **a**, and shows the P enrichment in natural marine barite

suggestion that in natural ocean water the bacterial biofilms would provide phosphate groups as binding sites for barite formation.

The bioaccumulation of Ba in the analyzed bacterial EPS would also suggest that barite formation is not associated with a specific organism as EPS is produced by an ample range of aquatic microorganisms, particularly phytoplankton and bacteria. This finding therefore opens a novel field surrounding the role of transparent exopolymer particles (TEPs) in the binding of Ba. TEPs play a major role in marine biogeochemical cycles and have received considerable attention since they were first described in the ocean[30,31]. They are ubiquitous in the aquatic system and drive the downward flux of POC, contributing to particle aggregation and the formation of the sinking marine snow[32]. Indeed, the bioconcentration of Ba in phytoplankton shown through culture and Ba isotope-spiking experiments[18] also suggests that EPS produced by phytoplankton could be similarly significant for barite precipitation mechanisms in the ocean, particularly because this production is quantitatively important. The binding of certain metals (e.g., Th) to functional groups of marine EPS (carboxylate, phosphate, and sulfate) has also been demonstrated[33] implying that a similar process may be prevalent for diverse authigenic minerals. Notably, the role of the EPS in the formation of other minerals (carbonates) has also been demonstrated even in the absence of living cells[34].

Crystallization of barite in the presence of organic compounds is also consistent with the typical rounded and elliptical barite morphologies found in seawater and marine sediments[35,36]. Additional evidence supporting biological mediation includes the distribution of Ba isotopes in seawater and particulate barites[12–14].

Based on all the above observations, we suggest that in regions where productivity is high and large amounts of organic matter are exported to depth and undergo bacterial degradation, EPS are commonly produced by the abundant phytoplankton and bacterial cells[37,38] and serve to bind the necessary Ba and form nucleation sites, leading to barite precipitation. This finding is consistent with the observed relationship between export productivity, bacterial activity, and barite formation in the ocean. Although additional research is needed to precisely describe the role of EPS and other biofilms in barite crystal nucleation and growth in the water column, our experimental work reveals a mechanism by which Ba bioaccumulation on microbial biofilms, and specifically in EPS, through phosphate binding groups would serve as the nucleation foci leading to barite formation in the ocean water column.

## Methods

**Culture media.** Bacterial strains were cultured and maintained in a marine broth (MB) medium (DIFCO laboratories, USA), with the following composition (in g/l): NaCl, 19.45; MgCl$_2$, 8.8; peptone, 5; Na$_2$SO$_3$, 3.24; CaCl$_2$, 1.8; yeast extract, 1; KCl, 0.55; NaHCO$_3$, 0.16; ferric citrate, 0.1; KBr, 0.08; SrCl$_2$, 0.03; H$_3$BO$_3$, 0.02; Na$_2$HPO$_4$, 8; Na$_2$SiO$_3$, 4; NaF, 2.4; NH$_4$NO$_3$, 1.6. The medium was used in liquid and solid forms; to prepare the solid medium, 2% purified agar-agar (OXOID laboratories, England) was added. For the Ba bioaccumulation experiments, sulfate-free synthetic seawater (SSW) was used to avoid inorganic barite precipitation. The composition of the SSW was (in g/l): NaCl, 19.45; MgCl$_2$, 5.9; CaCl$_2$, 1.8; KCl, 0.55; NaHCO$_3$, 0.16; KBr, 0.08; SrCl$_2$, 0.03; H$_3$BO$_3$, 0.02; ferric citrate, 0.1; Na$_2$HPO$_4$, 0.008; Na$_2$SiO$_3$, 0.004; NaF, 0.002; NH$_4$NO$_3$, 0.001; and pH 7.6. This SSW was used with two concentrations of BaCl$_2$: 2 mM (SSW1) and 20 mM (SSW2), both being higher than those found naturally in seawater. These concentrations served to expedite the process and allow for easy detection of the Ba bioaccumulation under the experimental conditions.

**Ba accumulation tests.** Bacterial cells were grown and maintained in MB medium. Bacterial growth was monitored in the liquid MB by measuring the optical density

at 620 nm. Ba accumulation tests were performed using bacterial biofilms produced by each of the selected strains. Cultures destined to biofilm formation were grown in liquid MB medium (100 ml) in 250 ml sterile Erlenmeyer flasks, incubated at 28 °C, and shaken continuously at a speed of 180 r.p.m. Cells were grown to late exponential phase and aliquots of 10 ml of the bacterial cultures were centrifuged at 10,000 r.p.m. for 10 min at 4 °C. The collected cells were rinsed three times with NaCl solution (3.5% w/v) to remove ingredients from the medium, in particular sulfates. The cells were re-suspended in sterile test tubes containing 10 ml of either SSW1 or SSW2. To promote biofilm formation, pieces of glass coverslips sterilized by autoclave (for 20 min at 120 °C) were aseptically placed in these previously prepared liquid-bacterial suspensions (one per each test tube). A minimum of three replicates per run were performed. The test tubes were incubated for 24, 48, and 96 h at 28 °C under shaking at 180 r.p.m.

**Electron microscopy analyses**. SEM and HRTEM were used to analyze the Ba bioaccumulated on bacterial biofilms. For this purpose, the glass coverslips were aseptically collected from the test tubes and immediately fixed in 1 ml of 2.5% glutaraldehyde solution (prepared in 0.2 M cacodylate buffer with 0.4 M sucrose and 0.1% NaCl to reach an osmolarity of 1220 mOsm similar to that of MB and SSW media). Samples were kept at 4 °C until their preparation for electron microscopy analyses. The fixed glass coverslips were dehydrated using a series of ethanol rinses (at 50; 70; 90; 100 vol% 3 ×; 10 min each), exposed to critical-point drying (in LEICA EM CPD 300 Critical Point Dryer) and coated with carbon prior to observations. The biofilms were characterized using SEM (AURIGA FIB-FESEM Carl Zeiss SMT and Zeiss SUPRA40VP), equipped with EDX detector system (Centre for Scientific Instrumentation, University of Granada). Biofilms were also analyzed by means of HRTEM; for these analyses, the coverslip surfaces with biofilms were scraped with a scalpel and then dispersed in ethanol by sonication for ~3 min. Samples were then collected and deposited on carbon-film-coated copper grids for observation. Data were collected using a FEI TITAN G2 60-300 microscope with a high brightness electron gun (X-FEG) operated at 300 kV and equipped with a Cs image corrector (CEOS). For analytical electron micro-scopy (AEM), a SUPER-X silicon-drift windowless EDX detector was used. Digital X-ray maps were collected on selected areas of the samples and mapped for Ba, S, and P. SAED patterns were also acquired for identification of crystalline phases. Regarding the analyses of barite particles from the ocean water column, representative pieces of the filters were coated with carbon for SEM observation.

**Data availability**. The datasets generated during the current study are available from the corresponding author.

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

## Acknowledgements

This study was supported by the European Regional Development Fund (ERDF) co-financed grant CGL2015-66830-R (MINECO Secretaría de Estado de Investigación, Desarrollo e Innovación, Spain), Research Groups BIO 103 and RNM-179 (Junta de

Andalucía), and the University of Granada (Unidad Científica de Excelencia UCE-PP2016-05).

## Author contributions

F.M.-R. and M.T.G.-M. jointly conceived and led this project and have written the manuscript. M.T.G.-M. also designed the experimental setup and F.J. conducted the experiments. A.P. contributed by discussing the results and writing and also provided marine barite samples and corresponding information, and I.G.-T. and M.d.M.A. contributed SEM and TEM analytical data, respectively.

## Additional information

**Competing interests:** The authors declare no competing interests.

