## [Peer Review File · Nature Communications]

Reviewers' comments:

Reviewer #1 (Remarks to the Author):

The manuscript is well written and the ideas are clearly presented in a logical fashion. The data appear sound and the arguments presented follow on from the available data.

The main problems I have with this manuscript is trying to decide what is new and what idea it presents that we did not know about before. The notion that biofilms/microbes may be instrumental in the accumulation of Ba in ocean setting is not in itself a new idea. Thus, I think that the authors need to clearly articulate and stress what is new and this differs from previous studies. I am not faulting the science here - I am simply asking that the "new factors" be clearly identified.

Reviewer #2 (Remarks to the Author):

Overall, I found the manuscript to be novel, interesting, informative, and of excellent quality. In my opinion, the manuscript should be published in Nature Communications. However, the following comments should be addressed to improve the manuscript rigor and quality:

- Several of the particles have substantial sulfur and iron signals. Could these features be iron sulfide minerals that precipitate in the biofilm and then adsorb barium? Do you also have element maps of iron and sulfur? Also, what is the source of the aluminum signal in the EDS spectra?
 - Please include "background" EDS spectra of the cover slip for comparison with the cell signature. It is not uncommon to get EDS signal from the mounting substrate, so it is important to show what the background looks like.
 - Did the authors do any sort of killed-cell control to determine whether or not active cells are required for barium accumulation? I'm not requesting it as a precondition for manuscript acceptance, but adding those data would be welcome if they have them.
 - What about testing the EPS alone – separate from the cells, to see if the cells or the EPS are most relevant for barium accumulation? Again, I'm not requesting it for manuscript acceptance, but adding those data would be welcome if they have them, or as follow-on experiments to consider for the future.
 - Line 90: includes a redundant "water" after seawater
 - . Line 113: This sentence could be worded more clearly. How about something like: "As was observed in our experiments, in the environmental samples, P was also localized with Ba, particularly in tiny grains (Fig. 114 4).
 - . Line 169: Did you do any DAPI staining to see if these cells make polyphosphate?
 - Line 178: the word "become" is extraneous in this sentence.
 - Line 178. Either the word "moreover" is being used incorrectly here, and the previous sentence is the supporting evidence for the claim "the composition of natural barites formed in the water column further support that a P-rich precursor is a mechanism for Ba binding and barite formation" made here or the claim is lacking a discussion of the evidence that supports it. This sentence and paragraph could use some reworking for clarity.
 - The screen captures of the EDS spectra are hard to read and won't reproduce well in print. I recommend recreating them as vector images or converting them to black and white bitmap traces with new text labels added.
- Otherwise looks good.
- Signed Jake Bailey

Reviewer #3 (Remarks to the Author):

Review of Martinez Ruiz et al, Barite bioaccumulation.....

The manuscript reports some elegant experimental and field results that suggest that bacterial mediated mechanism for the formation of barite xls. In particular the manuscript elucidates the role of EPS in bioaccumulating barite to promote barite precipitation in under saturated sea water. The results are important because barium is used extensively as a productivity indicator in the ocean and recently Ba isotopes have gained significant attention. The biomechanism espoused in the paper can explain the higher fractionation Ba isotopes in these studies than the predicted inorganic precipitation process.

In reading through the paper, the paper somewhat over plays the role of bacteria in this context. For instance, it ignores the role of phytoplankton. Most phytoplankton produce exopolymeric substances in significant quantities. For instance, this is how diatoms form clumps and sink in the ocean and it is essential part of diatoms life cycle. As a result phytoplankton themselves can bioaccumulate barium adsorb divalent cations on EPS and cell walls which in turns facilitate barite production while aggregation, sinking and decay. Quantitatively EPS produced by phytoplankton should be far greater than bacterial EPS production and hence more important as a Ba concentrating/barite precipitation mechanism in the ocean. The aspect is ignored in the manuscript.

The discussion on the role of the importance of bacteria 117-130 based on marine field studies- This are presented as direct evidence for importance of the role of bacterial in the bioaccumulation and production of barite. In fact there is ample evidence based on the same set of reference that abundance of barite xls in intermediate waters follow export (new) production at the surface. Thus, the barite production bound to follows production and sinking of phytoplankton biomass from the surface. Given that the decay of phytoplankton is bacterially mediated and bacterial abundance is likely to follow the availability substrate (see for instance Reference 30 and reference there in). This is not necessarily evidence for direct role of bacteria. The question is as a bioaccumulating mechanism whether the phytoplankton or the bacteria is more important in producing barite xls. The references here do not provide any evidence that the latter is more important given the emphasis on this process.

The authors rightly point out that bioconcentration of Ba is important in the sulphate-rich seawater (165-170). This has been specifically shown through Ba isotope spiking experiments in the context of phytoplankton production of barite (see Ganeshram et al., 2003, *Geochim. Cosmochim. Acta*, 67, 2599-2605). The author suggestion of intercellular barite production through initial Ba binding to polyphosphate granules within cells and subsequent replacement of phosphate with sulphate is a novel mechanism. The authors provide evidence for the association of P and Ba in their experimental studies.

In summary, the manuscripts some novel and important results that in addition to Phytoplankton bacteria themselves can play a role in bioconcentrating Ba and promote barite production in the marine environment. I think the manuscript will benefit from rewriting reflecting more this nuance of barite production in the ocean and placing the role of bacteria in context of other mechanism such as phytoplankton.

Finally, this manuscript (abstract and results) is remarkably similar to a previous paper. Eg. reference 13 in the manuscript. It is important that the authors highlight new results and how this manuscript adds significantly to the published manuscript before warranting publication in *Nature Comm.*

Reviewer #1

The main point raised by this reviewer is the novelty of the notion that biofilms/microbes may be instrumental in the accumulation of Ba in ocean setting. We fully agree with their comment and indeed some of the papers that first reported Ba enrichments in marine sediments around the early sixties (e.g., Goldberg and Arrhenius, 1958; Chow and Goldberg, 1960) reported the link between barite formation in the ocean and biological activity, and Church already proposed in 1970 that Ba is extracted from seawater by marine organisms. Later on Bishop et al. (1988) stated that “barite formation appears to be caused by biological activity in the upper water column but the exact mechanism is unknown” and their results indicated first that “barites are formed almost exclusively in microenvironments containing decaying organic matter and the remains of siliceous plankton” and second that “barites do not appear to be actively formed by the planktonic organisms sampled”. Nevertheless, in spite of decades of research, the biogeochemistry of barium has intrigued geochemists for decades because the precise mechanism for the precipitation of Ba as barite in undersaturated conditions is still unknown. Although a clear association to organic matter production and carbon fluxes has been demonstrated, the exact mechanisms for nucleation and crystallization are not understood.

In this regard, our group started to investigate the potential microbial role in barite precipitation in 2000. Most of the research on mineral bacterial precipitation had been focused on carbonates and iron minerals but little had been done on sulfates by that time. As an initial approach, a soil bacterium with a well-known biomineralization capability (*Myxococcus xanthus*) was used for our experiments and we demonstrated for the first time that bacteria could mediate the precipitation of barite under experimental conditions. After that, Glamoclija et al. (2004), Sanchez-Moral et al. (2004) and Senko et al. (2004) also reported barite precipitation in natural environments where bacteria may have played a role either oxidizing sulfur compounds to generate sulfate or providing biofilms favoring biomineralization. Also in a warm sulfur spring in Canada, Bonny and Jones (2008) reported barite crystals that nucleated on microbial cell surfaces and in microbial extracellular polymeric substances (EPS). However, the marine environment had not been investigated in this sense. As a further step in our research in this field, we carried our diverse experiments with marine bacteria and demonstrated that several strains have the capability to precipitate barite under laboratory conditions (González-Muñoz et al. 2012), which opened an exciting field of research to investigate the bacterial role in natural environments and interestingly supports the potential bacterial role in barite precipitation in the ocean water column. It has also been shown that higher mesopelagic Ba in high productivity regions in the ocean correlates with greater bacterial activity, suggesting a potential relationship (Dehairs et al., 2008; Jacquet et al. 2011; Planchon et al., 2013). Herewith, our new approach is therefore to provide new insights from experiments in conditions closer to the natural environment, where bacteria are not in culture petri dishes but forming biofilms. Moreover in our previous experiments the bacterial metabolism was responsible for providing the necessary sulfur for Ba precipitation, however in the ocean sulfate is readily available, and precipitation in undersaturated seawater strongly depends on Ba concentration and saturation. This is the process that is still unknown, and aiming to further understand this process we designed laboratory experiments in

which the bacterial role was not providing sulfate but concentrating Ba and also in which bacteria growth was more similar to the natural occurrence of bacteria in seawater. In this context, our results demonstrate the capability of bacterial biofilms to concentrate Ba leading to saturation and barite formation. In summary, the experimental work and results included in this manuscript differ from the previous one in:

- The experimental setup was designed to detect biofilms since these represent the natural occurrence of bacteria in natural environments.
- The objective that is to demonstrate the capability of biofilms in concentrating Ba rather than providing the sulfur (through metabolic processes) as was in previous culture experiments.
- The role of EPS in concentrating Ba had been suggested but not demonstrated and definitely not in seawater.
- The focus on nucleation and crystallization of Ba mineral phases, though a P-rich precursor had been previously shown.

These aspects have been clarified in the revised manuscript and the information concerning novel aspects has been included in the text (see lines 26-31 in the abstract and following sections 147-148, 182-184). Specifically, we emphasize that this is the first time that a direct link between biofilms and concentrating barite under conditions more like seawater and using marine bacteria. This is also the first time that precursor barite crystals have been detected in natural marine samples (see lines 115-121).

Reviewer #2

We really appreciate the positive opinion of the reviewer on our manuscript and particularly his constructive comments that notably help to improve the original version. The response to his questions and points raised is provided below (cursive is used for the paragraphs/sentences taken from the review):

- *Several of the particles have substantial sulfur and iron signals. Could these features be iron sulfide minerals that precipitate in the biofilm and then adsorb barium? Do you also have element maps of iron and sulfur? Also, what is the source of the aluminum signal in the EDS spectra?*
- *Please include "background" EDS spectra of the cover slip for comparison with the cell signature. It is not uncommon to get EDS signal from the mounting substrate, so it is important to show what the background looks like.*

The iron sulphide mineral phase would be easily identified under the SEM in back scattered mode and no iron sulphides have been observed. Additionally the iron is present in the biofilms prior to sulfur being concentrated, which suggest that iron is also concentrated in the biofilms just like Ba. In general biofilms have the capability to concentrate diverse metals, and this has been extensively demonstrated (see lines 194-196), thus we interpret the iron enrichment in a similar way as Ba and it is not linked to sulphides formation. Iron is present as ferric citrate, 0.1 g/L, in the synthetic seawater used for the experiments, hence it will be bound to the EPS like any other cation. Moreover, the bacteria used in the experiments are not sulfate-reducing microorganisms and conditions of the medium are not reducing (e.g., the culture is oxic).

A representative example of a map of iron and sulfur is provided below and no correlation is observed. While sulfur is generally distributed in bacterial cells (e.g., as part of proteins) no clear enrichment in iron is observed linked to sulfur.

Fig. R1. HAADF images obtained by HRTEM showing bacteria from the biofilms (*I. loihiensis* MAH1) and EDX maps showing the S and Fe distribution.

Regarding the Al signal, background spectra are provided in the supplementary material and the information is also provided in the result section (see lines 94-97). We fully agree with this point since no information of the background elements was provided, thus it was not clear which one are enriched in biofilms and which are from the glass cover slips used for the experiments. In this case Al, and other elements as K and abundant Si are derived from that glass.

- Did the authors do any sort of killed-cell control to determine whether or not active cells are required for barium accumulation? I'm not requesting it as a precondition for manuscript acceptance, but adding those data would be welcome if they have them.

We did not include a killed control in our experiments because the goal was to show that bacteria form biofilms that in turn adsorb the Ba. If killed obtaining biofilms would not be possible. Regarding this aspect, there is an interesting paper that demonstrates that living cells are not always required for microbial mineral precipitation in the case of carbonates, but they precipitate in the EPS (Bontognali et al., 2012). We refer to this paper (see lines 200-203) and suggest that this is also likely to be the case for barite.

Specifically, we agree with the reviewer and also think that even in the absence of living cells the Ba could be concentrated if EPS is present. Indeed, our work emphasizes the important role of the EPS in the precipitation (with or without living cells).

- What about testing the EPS alone – separate from the cells, to see if the cells or the EPS are most relevant for barium accumulation? Again, I'm not requesting it for manuscript

acceptance, but adding those data would be welcome if they have them, or as follow-on experiments to consider for the future.

We fully agree with this idea, separating EPS from the cells would be interesting in order to explore the role of each in concentrating Ba. However, we have not done this in our study because in seawater bacteria commonly occur with and in biofilms. Additionally, we have observed Ba enrichment in both, cells and EPS, so it seems like there are two processes occurring. Nevertheless, this is a very interesting point for future research and future experiments. In our opinion, not only in carbonate precipitation, as reported in the paper mentioned above, but also in the precipitation of clay minerals or barite the EPS role is crucial and live bacteria are not needed as long as there is a source of EPS.

- Line 90: includes a redundant "water" after seawater

- Line 113: This sentence could be worded more clearly

These sentences have been modified following the reviewer suggestion (see lines 90 and 115-117).

- Line 169: Did you do any DAPI staining to see if these cells make polyphosphate?

No, the DAPI staining was not done because our main focus has been the EDX analyses of Ba precipitates. However, the morphology and composition of the granule shown in Fig. 3 is indicative of a polyphosphate (see spectrum below). Additionally, it has been previously demonstrated that polyphosphate granules of Myxobacterias can absorb Ba as Barium phosphate (Merroun, 1999). As also observed in our spectra (see Fig. R2 below), polyphosphate granules are particularly rich in P and Ca (e. g., Fig. 2 from Krawczyk-Bärsch et al., 2018). Furthermore, when comparing the intensity of the P peak in the spectra obtained in these granules, it is higher than that corresponding to Ba accumulated in the EPS (see HRTEM spectrum included in the supplementary material) supporting our suggestion that the granules are polyphosphate. Thus, we claim that binding of Ba to polyphosphate is a common process in bacterial biofilms. Nevertheless, as we did not use DAPI staining we only suggest in the manuscript that these granules are **likely** polyphosphate granules. This is also a very exciting aspect and it will be interesting to design future experiments to demonstrate the production of polyphosphate granules and also use DAPI staining. Reviewer #3 also indicates the importance of the potential link of polyphosphate granules with barite formation.

Fig. R2. HRTEM spectrum showing the composition of the likely polyphosphate granule shown in Fig. 3.

- Line 178: the word "become" is extraneous in this sentence.
- Line 178. Either the word "moreover" is being used incorrectly here...

The sentences have been corrected following the reviewer suggestion (see new paragraph, lines 205-213).

- The screen captures of the EDS spectra are hard to read and won't reproduce well in print. I recommend recreating them as vector images or converting them to black and white bitmap traces with new text labels added.

This has been done; see new spectra in the figures.

Reviewer #3

We agree with the reviewer and acknowledge the valuable comments that also contribute to improve the revised version and add a new perspective to the manuscript on other potential Ba bioaccumulation mechanisms in the ocean water column. In summary, the reviewer points to the role of bacteria that has been over reflected in the

manuscript against that of phytoplankton. This has been considered in the revised version and new paragraphs to discuss this have been added:

- See lines 74-77 regarding phytoplankton production in the introduction, this new perspective has been considered.
- See lines 138-142 on phytoplankton bioaccumulating barium in the discussion section.
- The discussion has been extended (see lines 186-213) regarding barite production following productivity and sinking of phytoplankton biomass from the surface as well as highlighting the importance of phytoplankton and EPS production.

The reviewer also considers that the manuscript is remarkably similar to a previous paper (reference 13 in previous version), but this has been explained above in response to reviewer #1 and novel aspects have been highlighted in the abstract and throughout the manuscript.

References

- Bishop, J.K.B. The barite–opal–organic carbon association in oceanic particulate matter *Nature*, 332, 341-343(1988).
- Bonny, S.M. & Jones, B. Diatom-mediated barite precipitation in microbial mats calcifying at Stinking Springs, a warm sulphur spring system in Northwestern Utah, USA. *Sediment. Geol.* **194(3)**, 223-244 (2007).
- Bontognali, T.R.R., Mckenzie, J.A., Warthmann, R.J. & Vasconcelos, C., Microbially influenced formation of Mg-calcite and Ca-dolomite in the presence of exopolymeric substances produced by sulphate-reducing bacteria. *Terra Nova*, **26 (1)**, 72-77 (2014).
- Chow, J.& Goldberg, E.D. On the marine geochemistry of barium. *Geochim. Cosmochim. Acta* **20** (3–4), 192-198 (1960).
- Church T.M. Marine barite (third ed.), Ph. D. Thesis, Univ. of California, San Diego (1970).
- Dehairs, F. et al. Barium in twilight zone suspended matter as a potential proxy for particulate organic carbon remineralization: Results for the North Pacific. *Deep Sea Res. Part II Top. Stud. Oceanogr.* 55(14), 1673-1683 (2008).
- Glamoclija, M., Garrel, L., Berthon, J. & López-García, P. Biosignatures and bacterial diversity in hydrothermal deposits of Solfatara Crater, Italy. *Geomicrobiol. J.* **21(8)**, 529-541 (2004).
- Goldberg, E.D. & Arrhenius, G. Chemistry of pelagic sediments. *Geochim. Cosmochim. Acta* **13**, 153–212 (1958).
- González-Muñoz, M.T., Martínez-Ruiz, F., Morcillo, F., Martín-Ramos, J.D. & Paytan, A. Precipitation of barite by marine bacteria: A possible mechanism for marine barite formation. *Geology* **40(8)**, 675-678 (2012).

Jacquet, S.H. et al. Twilight zone organic carbon remineralization in the Polar Front Zone and Subantarctic Zone south of Tasmania. *Deep Sea Res. Part II Top. Stud. Oceanogr.* **58(21)**, 2222-2234 (2011).

Krawczyk-Bärsch, E., Gerber, U., Müller K., Moll, H., Rossberg, A., Steudtner, R. & Merroun, M.L. Multidisciplinary characterization of U(VI) sequestration by *Acidovorax facilis* for bioremediation purposes. *J. Hazard. Mater.* **347**, 233-241 (2018).

Merroun, M. L., 1999. Biosorption of metals by *Myxococcus xanthus*: *PhD Thesis*. University of Granada, Spain.

Planchon, F., Cavagna, A.J., Cardinal, D., André, L. & Dehairs, F. Late summer particulate organic carbon export and twilight zone remineralisation in the Atlantic sector of the Southern Ocean. *Biogeosciences* **10**, 803-820 (2013).

Sánchez-Moral, S. et al. Bioinduced barium precipitation in St. Callixtus and Domitilla catacombs. *Ann. Microbiol.* **54(1)**, 1-12 (2004).

Senko, J.M., Campbell, B.S., Henriksen, J.R., Elshahed, M.S., Dewers, T.A. & Krumholz, L.R. Barite deposition resulting from phototrophic sulfide-oxidizing bacterial activity: *Geochim. Cosmochim. Acta.* **68**, 773-780 (2004).

REVIEWERS' COMMENTS:

Reviewer #1 (Remarks to the Author):

I have now reviewed the revised version of your manuscript while considering ALL of the comments that offered in reply to the issues that were raised with respect to the original version of this manuscript. In my opinion, the manuscript has been much improved and all of the comments offered by the reviewers have been dealt with in a very positive manner. Thus, I would recommend publication.

Reviewer #2 (Remarks to the Author):

I am satisfied with the author's responses to the review comments and recommend publication.